# Rethinking Graph Attention Networks: A New Robust Approach

## Abstract

Graph Attention Networks (GAT) have achieved remarkable success for representation learning with graphs. However, their performance is significantly degraded due to the oversmoothing in which deep GAT leads to homogenized node representations. In this paper, we introduce a new quantitative measure of oversmoothing based on Mahalanobis distance. This measure provides a more robust assessment than conventional Euclidean metrics. Based on this insight, we propose the Mahalanobis Graph Attention Network (MGAT) to alleviate the oversmoothing issue. MGAT adds a Mahalanobis regularizer to reduce representation collapse and preserves inter-class separability. Extensive experiments on common benchmark datasets demonstrate the efficiency and superiority of our proposed model compared to the base GATs.

## 1 Introduction

Graph Neural Networks (GNNs) Ye et al. (2022); Wu et al. (2020); Scarselli et al. (2009); Cai & Wang (2020) have emerged as a powerful paradigm for representation learning on graph-structured data. By leveraging message passing and neighborhood aggregation, GNNs are able to capture both local and global structural dependencies in graphs, which has led to successful applications in social network analysis, recommender systems, biological networks, and knowledge graphs. Early models such as Graph Convolutional Networks (GCNs) David Kristjanson Duvenaud & Adams (2015); Michaël Defferrard & Vandergheynst (2016) demonstrated that spectral graph convolution provides an effective way to generalize convolutional neural networks to irregular graph domains. However, GCNs rely on fixed aggregation weights derived from the graph Laplacian, which limits their flexibility in handling heterogeneous neighborhood information.

To overcome this limitation, Graph Attention Networks (GATs) Veličković et al. (2018) introduced attention mechanisms into graph learning, allowing the model to assign adaptive importance scores to neighbors during message passing. This innovation enables GATs to focus on the most relevant neighbors while downweighting less informative ones, making them more expressive and versatile compared to GCNs and other earlier architectures. As a result, GATs have become one of the most widely adopted GNN variants in recent years.

Nevertheless, despite their empirical success, GATs still suffer from the fundamental limitation of *oversmoothing* Zhao & Akoglu (2008); Xinyi Wu & Jadbabaie. (2023). As the number of layers increases, repeated propagation causes node embeddings to lose diversity and gradually converge to indistinguishable representations within the same connected component Xiaojun Guo & Wang (2023). This collapse severely reduces discriminative power for downstream tasks such as node classification. While attention mechanisms were expected to mitigate this issue by differentiating neighbor contributions, recent studies indicate that deep GATs remain highly vulnerable to oversmoothing. This leads to several critical questions: To what extent can attention truly prevent oversmoothing in practice? Which feature-space directions are most prone to collapse? And can we design principled methods to quantify and explicitly mitigate oversmoothing in GATs, rather than relying solely on heuristic modifications?

From a mathematical perspective, oversmoothing in GATs can be explained through the repeated application of attention-weighted propagation. A generic GAT layer can be written as $X^{(\ell+1)} = \sigma(P^{(\ell)} X^{(\ell)} W^{(\ell)})$, where $P^{(\ell)}$ is the attention-based propagation matrix. After stacking $T$ layers,

the embeddings become $X^{(T)} \approx (\prod_{\ell=0}^{T-1} P^{(\ell)}) X^{(0)} W$. Under mild assumptions on the propagation matrices, this product tends to project embeddings onto a subspace close to the kernel of the normalized Laplacian $\ker(L)$ $L = I - D^{-1/2} A D^{-1/2}$. Since $\ker(L)$ Singer (2006) is spanned by constant vectors, embeddings across connected nodes become nearly identical, thereby collapsing useful variability in the features. Classical metrics such as the Dirichlet energy $\mu(X) = \text{tr}(X^\top L X)$ capture this phenomenon by measuring how much embeddings vary along graph edges. However, such Euclidean-based measures treat all feature directions equally, and therefore cannot distinguish between collapse in discriminative dimensions versus collapse in less relevant ones.

To address this shortcoming, we propose a Mahalanobis-based formulation of energy De Maesschalck et al. (2000). Specifically, given a positive semidefinite matrix $M \succeq 0$, we define $\mu_M(X) = \sqrt{\text{tr}(X^\top L X M)}$. Unlike the Euclidean case, where all directions are penalized equally, the Mahalanobis metric introduces anisotropy into the assessment of smoothness. If $M$ admits eigenvalues $\{\lambda_k\}$ with eigenvectors $\{v_k\}$, then $\mu_M(X) = \sum_k \lambda_k \|L^{1/2} X v_k\|_2^2$, showing that directions associated with larger eigenvalues are penalized more strongly, while directions with smaller eigenvalues are preserved Hoerl & Kennard (1970); Hastie & Friedman (2009). Thus, Mahalanobis energy enables adaptive smoothing control: irrelevant or noisy components can be smoothed aggressively, whereas informative components are protected. Moreover, the gradient of this energy with respect to embeddings is proportional to $LXM$, which actively counteracts convergence into $\ker(L)$ Singer (2006) unless the embeddings lie simultaneously in the null spaces of both $L$ and $M$. This property gives Mahalanobis regularization a strong theoretical basis for maintaining inter-class separability and mitigating oversmoothing in deep GATs.

Building on these insights, we propose the Mahalanobis Graph Attention Network (MGAT), which integrates a Mahalanobis regularizer into the learning objective. This framework explicitly controls oversmoothing during training, improving robustness to network depth while maintaining flexibility in neighbor aggregation. Extensive experiments on benchmark datasets validate the effectiveness of MGAT, demonstrating superior performance compared to standard GATs and confirming that our Mahalanobis-based approach provides both practical and theoretically grounded benefits.

## 2 RELATED WORK

Oversmoothing Han Shi & Kwok (2022); Keriven (2022) has been recognized as a fundamental challenge in deep graph neural networks. Initial studies in GCNs Cai & Wang (2020) showed that as the number of layers increases, node representations tend to converge, leading to diminished performance on downstream tasks. Researchers have proposed a variety of techniques to mitigate this phenomenon, such as residual connections, skip connections, and normalization methods. Nevertheless, oversmoothing remains a key bottleneck for scaling GNNs to deeper architectures.

GATs Shaked Brody & Yahav (2022) introduced attention mechanisms to graph learning, allowing networks to learn which neighbors are more important during message aggregation. While attention helps emphasize informative nodes, empirical studies have shown that deep GATs can still experience representation collapse, particularly in large or densely connected graphs. This observation motivates the need for methods that go beyond attention alone to explicitly address oversmoothing.

Most prior works assess oversmoothing using average node similarity or Euclidean distance metrics Kaixiong Zhou & Hu (2021); Rusch et al. (2023); Cai & Wang (2020). These measures do not distinguish between meaningful variations in discriminative features and uniform collapse across less relevant directions. This limitation highlights the need for a more principled, direction-sensitive metric for assessing oversmoothing.

Several approaches have been proposed to improve GNN expressiveness and combat oversmoothing, including edge dropout, feature decorrelation, and label propagation strategies. These methods are generally heuristic and do not provide explicit control over which feature directions are smoothed. In contrast, our proposed MGAT introduces a Mahalanobis-based regularizer De Maesschalck et al. (2000) that adaptively penalizes oversmoothing along specific directions in the feature space, preserving discriminative information and improving the network's capacity to learn expressive node representations.

The main contributions of this work can be summarized as follows:

1. A novel measure of oversmoothing: We introduce a Mahalanobis-based metric that quantifies oversmoothing in GATs more robustly than conventional Euclidean metrics, allowing for direction-sensitive assessment of representation collapse.

2. Mahalanobis Graph Attention Network (MGAT): Based on this measure, we propose MGAT, which incorporates a Mahalanobis regularizer to mitigate oversmoothing. This framework adaptively preserves discriminative feature directions while controlling smoothing across layers.

3. Empirical validation: We conduct extensive experiments on standard benchmark datasets, demonstrating that MGAT consistently outperforms baseline GATs in node classification and robustness, validating the effectiveness of our proposed metric and regularization strategy.

The rest of this paper is organized as follows. Preliminaries is described in section 3. Section 4 shows our main results. Section 5 provides experiments and finally, section 6 concludes the paper.

## 3 PRELIMINARIES

A graph neural network (GNN) layer updates each node's representations by aggregating information from its neighbors.

Let $G = (V, E)$ be an undirected graph, where $V = \{v_1, \ldots, v_N\}$ is the set of $N$ nodes and $E \subseteq V \times V$ is the set of edges. The adjacency matrix of $G$ is denoted by $A \in \mathbb{R}^{N \times N}$. $D \in \mathbb{R}^{N \times N}$ is the degree matrix ($D_{ii} = \sum_j A_{ij}$). Each node $v_i$ is associated with a representational vector $h_i \in \mathbb{R}^C$, and the collection of all node representations is represented as

$$X = \{h_1, h_2, \ldots, h_N\} \in \mathbb{R}^{N \times C},$$

A single GNN layer computes updated node representations. In the output layer, the updated representation set is given by

$$X' = \{h'_1, h'_2, \ldots, h'_N\}, \quad h'_i \in \mathbb{R}^{C'},$$

and computed as

$$h'_i = f_\theta \Big( h_i, \text{AGGREGATE}\{h_j \mid j \in \mathcal{N}_i\} \Big),$$

where $\mathcal{N}_i = \{j \in V \mid (j, i) \in E\}$ is the set of neighbors of $v_i$, $C'$ is the dimensionality of the output representations, and $f_\theta$ denotes a learnable transformation parameterized by $\theta$.

The dimension $C'$ may differ from $C$ because the GNN layer not only combines neighborhood information but also transforms it into a new representation space, enabling the network to capture patterns more effectively for the target task.

### 3.1 GRAPH ATTENTION NETWORKS

The question behind the representation learning approaches focuses on determining when two representations can be considered similar. Graph Attention Network (GAT) Veličković et al. (2018) operates by computing similarity and using that similarity to propagate information from neighboring nodes. Computing similarity involves a series of sequential operations: initializing similarity, strengthening similarity between closely related nodes or dissimilarity among weakly related nodes, and normalizing the resulting similarity matrix.

#### 3.1.1 INITIALIZING SIMILARITY:

In GATs, similarity is computed using a *binary function* $f : V \times V \to \mathbb{R}$, $f$ is metric-based similarity function below:

$$f(x, y) = -\sigma \left( \alpha_1^\top P x - \alpha_2^\top P y \right)$$

where $\alpha_i$ ($i = 1, 2$) are learnable vectors, $P$ is a learnable matrix, and $\sigma$ is an activation function. The specific pseudo-metric function of attention mechanism to calculate similarity between node $x_i$ and node $x_j$ is expressed as

$$-f(x_i, x_j) = \text{LeakyReLU}\left(\vec{a}^\top \left[W\vec{h}_i \,\|\, W\vec{h}_j\right]\right),$$

where $\|\cdot\|$ denotes vector concatenation, weight vector $\vec{a} \in \mathbb{R}^{2F'}$, and weight matrix $W \in \mathbb{R}^{C' \times C}$.

### 3.1.2 STRENGTHENING SIMILARITY:

The purpose of strengthening similarity is to make two data points that are relatively similar become even more similar and decreasing weaker ones to aid the aggregating process. The specific attention mechanism utilizes the exponential inflation.

*(Exponential Inflation). The exponential inflation operator $\Gamma$ is defined as follows*

$$\Gamma : \mathbb{M}_{N \times N} \to \mathbb{M}_{N \times N}$$

$$\Gamma(M)_{ij} = \exp(\frac{-f(x_i, x_j)}{\epsilon})$$

### 3.1.3 NORMALIZING SIMILARITY:

The purpose of normalizing similarity is to yield a probabilistic distribution suitable for comparative analysis or downstream processing.

*Row Normalization.* The row normalization operator $N_r : \mathbb{M}_{N \times N} \to \mathbb{M}_{N \times N}$ is expressed as follows:

$$P_{ij} = \frac{\exp(-f(x_i, x_j))}{\sum_{k \in \mathcal{N}} \exp(-f(x_i, x_k))}$$

### 3.1.4 INFORMATION PROPAGATION

The information propagation of a single graph attentional layer is given by:

$$X^{(t+1)} = \sigma(PX^{(t)}W)$$

## 4 MAIN RESULTS

### 4.1 OVERSMOOTHING

We formalize the concept of oversmoothing following the framework of Rusch et al. Rusch et al. (2023).

**Definition 4.1** (Node Similarity Measure). *Let $G = (V, E)$ be an undirected, connected graph with $|V| = N$, and let $\mu : \mathbb{R}^{N \times C} \to \mathbb{R}_{\geq 0}$ denote a function that quantifies the pairwise similarity of node representations. We call $\mu$ a node similarity measure if it satisfies:*

*1. There exists $c \in \mathbb{R}^C$ such that $X_i = c$ for all $i \in V$ if and only if $\mu(X) = 0$ for $X \in \mathbb{R}^{N \times C}$.*

*2. For all $X, Y \in \mathbb{R}^{N \times C}$, $\mu(X + Y) \leq \mu(X) + \mu(Y)$.*

Definition 4.1 formalizes the notion of a node similarity measure $\mu$, which is used to quantify how similar the feature representations of nodes are across a graph. The first condition states that there exists a vector $c \in \mathbb{R}^C$ such that $X_i = c$ for all nodes $i \in V$ if and only if $\mu(X) = 0$. Mathematically, this ensures that $\mu$ measures the *variation* among node embeddings. This property guarantees that $\mu$ reflects the essential intuition of similarity: zero measure corresponds exactly to perfect homogeneity of node features. The second condition is a subadditivity condition analogous to the triangle inequality in normed spaces. This property is crucial in analyzing oversmoothing because propagation operators in GATs add linear transformations and aggregations of node features.

**Definition 4.2** (Oversmoothing). *Given a node similarity measure $\mu$, we say that oversmoothing occurs if*

$$\lim_{t \to \infty} \mu\left(X^{(t)}\right) = 0,$$

*where $X^{(t)}$ denotes the matrix of node representations after $t$ layers.*

$\mu(X)$ measures the *pairwise dissimilarity* among node features. Mathematically, the definition 4.2 implies that as $t \to \infty$, the difference between any two node representations vanishes: $\forall i, j \in V$, $\lim_{t\to\infty} \|X_i^{(t)} - X_j^{(t)}\| = 0$. In other words, all node embeddings converge to a common vector, losing the ability to distinguish nodes based on their features in deep GATs.

**Mahalanobis-based Node Similarity.** We study oversmoothing in GATs using the following node similarity measure derived from the Mahalanobis distance:

$$\mu_M(X) = \left\| L^{1/2} X M^{1/2} \right\|_F,$$

where $L \in \mathbb{R}^{N \times N}$ is the Laplacian of a connected, undirected graph, $M \in \mathbb{R}^{C \times C}$ is positive definite, and $\| \cdot \|_F$ is the Frobenius norm

The graph Laplacian $L = D - A$, where $D$ is the degree matrix and $A$ is the adjacency matrix, encodes the connectivity structure of the graph.

$$(LX)_i = \sum_{j \in \mathcal{N}(i)} (X_i - X_j),$$

and thus $LX$ captures the smoothness of the feature matrix over the graph. The term $XM$ reweights the feature space so that differences along directions of higher variance are penalized less, while differences along directions of lower variance are amplified. Mathematically, it corresponds to the Mahalanobis distance:

$$d_M(x_i, x_j) = \sqrt{(x_i - x_j)^\top M (x_i - x_j)}.$$

**Proposition 4.3.** $\mu_M(X) = \left\| L^{1/2} X M^{1/2} \right\|_F$ *satisfies node similarity measure*

The measure $\mu_M(X)$ combines structural information from the graph Laplacian $L$ and feature correlations from $M$. It quantifies the overall dissimilarity between node features while respecting the graph topology.

**Theorem 4.4.** *Suppose there exist constants $\alpha \in [0, 1)$ and $\gamma > 0$ such that for all feature matrices $X$ occurring in the propagation,*

$$\left\| L^{1/2} P(X) L^{+1/2} \right\|_2 \leq \alpha \qquad \text{and} \qquad \|W\|_2 \leq \gamma,$$

*where $L^{+1/2}$ is the square root of the Moore-Penrose pseudoinverse of $L$. Then,*

$$\lim_{t\to\infty} \mu\big(X^{(t)}\big) = 0.$$

The condition $\alpha < 1$ ensures that $P(X)$ as a contraction in the node feature space with respect to the Laplacian-induced norm. The bound on $\|W\|_2$ ensures that the linear transformation does not amplify the features excessively. Together, these conditions guarantee that each propagation step brings node representations closer together rather than spreading them apart.

## 4.2 SOLUTION

Gradient descent on neural networks may converge to minimization of the loss. Without a regularizer, the model might converge to solutions with low margin, leading to poor generalization or collapsed embeddings Colin Wei & Ma (2019). In the context of Graph Neural Networks (GNNs), this manifests as *oversmoothing*.

**Definition 4.5** (Regularization Loss). *Add a regularizer $R(\theta)$ to the task loss:*

$$L(\theta) = L_{task}(\theta) + \lambda R(\theta),$$

*where $\lambda > 0$ balances the task performance and the effect of regularization.*

The paper Colin Wei & Ma (2019) shows that regularization helps gradient descent converge to global minima with larger margin. This ensures that embeddings are more distinct, preventing collapse into low-information subspaces. The total gradient of the regularized loss is

$$\nabla_\theta L = \nabla_\theta L_{\text{task}} + \lambda \nabla_\theta R(\theta).$$

At convergence, $\nabla_\theta L = 0$, The task loss gradient $\nabla_\theta L_{\text{task}}$ pulls the embeddings toward minimize the task loss, which could reduce embedding diversity. The regularizer gradient $\lambda \nabla_\theta R(\theta)$ pushes embeddings away from low-information configurations, preserving diversity.

**Mahalanobis-Graph Attention Networks (MGAT) Methodology.** Repeated propagation through multiple GNN layers causes node embeddings to converge toward the kernel of the graph Laplacian, resulting in oversmoothing. Inspired by the concept of regularization preserving embedding diversity, we propose MGAT methodology to mitigate oversmoothing in deep GATs.

To control smoothing adaptively along different feature directions, we define the Mahalanobis energy at layer $\ell$ as

$$\mu_M(X^{(\ell)}) = \|L^{1/2} X^{(\ell)} M^{1/2}\|_F.$$

This energy measures the smoothness of node embeddings along directions weighted by $M$. The gradient of this energy with respect to $X^{(\ell)}$ is

$$\nabla_{X^{(\ell)}} \mu_M(X^{(\ell)}) = \frac{L X^{(\ell)} M}{\|L^{1/2} X^{(\ell)} M^{1/2}\|_F}.$$

which prevents embeddings from collapsing into $\ker(L)$.

Suppose $M$ has eigenvectors $\{v_k\}$ with corresponding eigenvalues $\{\lambda_k\}$. Then the Mahalanobis energy can be expressed as

$$\mu_M(X) = \sum_k \lambda_k \|L^{1/2} X v_k\|_2^2.$$

Mahalanobis energy allows adaptive smoothing along directions with large eigenvalues of $M$ and preserving important directions with small eigenvalues Hoerl & Kennard (1970); Hastie & Friedman (2009). By learning $M$ during training to ensure positive semi-definiteness, the network automatically determines which directions to smooth and which to preserve, achieving anisotropic smoothing.

We incorporate the Mahalanobis energy as a regularizer in the overall loss:

$$\mathcal{L} = \mathcal{L}_{\text{task}} + \lambda \sum_{\ell=0}^{T-1} \mu_M(X^{(\ell)}), \quad \lambda > 0,$$

where $\mathcal{L}_{\text{task}}$ is the task-specific loss and $\lambda$ balances task performance with oversmoothing mitigation. During gradient-based optimization, the term $\nabla_{X^{(\ell)}} \mu_M(X^{(\ell)})$ ensures that embeddings are prevented from collapsing entirely into $\ker(L)$, while the learnable metric $M$ provides directional control, allowing the model to retain important features and mitigate oversmoothing without reducing expressive power.

## 5 EXPERIMENTS

In the experimental evaluation, we aim to address the following research questions:

1. **RQ1.** Does MGAT improve node classification accuracy compared to GAT on standard benchmark datasets?

2. **RQ2.** Can MGAT mitigate the effect of oversmoothing and maintain performance as the network depth increases? We do experiments for Mahalanobis energy and Dirichlet energy Kaixiong Zhou & Hu (2021); Rusch et al. (2023); Cai & Wang (2020) to answer this question.

   **Mahalanobis Energy.** $\mu_M(X^{(\ell)}) = \|L^{1/2} X^{(\ell)} M^{1/2}\|_F$, $L^{1/2} X^{(\ell)}$ measures differences of embeddings between neighboring nodes. Multiplying by $M^{1/2}$ allows the network to weight different feature dimensions differently, emphasizing informative directions and suppressing uninformative ones. reserving $\mu_M$ ensures that reprensentations do not collapse into similar vectors, which mitigates oversmoothing.

   **Dirichlet Energy.** $\mu_D(X^{(\ell)}) = \|L^{1/2} X^{(\ell)}\|_F$, which treats all feature dimensions equally and only enforces smoothness along edges.

3. **RQ3.** Compared to other state-of-the-art methods designed to alleviate oversmoothing, does MGAT achieve competitive or superior performance?

**Dataset.**    Joining the previous studies, we evaluate the performance of GNN models on node classification tasks using three widely-used real-world datasets: Cora, Citeseer, and Pubmed Sen et al. (2008). The detailed statistics of these datasets, along with the data splits, are summarized in Table 1

Table 1: Dataset statistics.

| Dataset | Nodes | Edges | Features | Classes |
|---------|-------|-------|----------|---------|
| Cora | 2,708 | 5,429 | 1,433 | 7 |
| Citeseer | 3,327 | 4,732 | 3,703 | 6 |
| Pubmed | 19,717 | 44,338 | 500 | 3 |

We compare MGAT against several representative and widely adopted baselines that address over-smoothing in different ways. Specifically, we include the original Graph Attention Network (GAT) Veličković et al. (2018), which employs attention mechanisms for neighbor aggregation, and DropEdge Y. Rong (2019), a stochastic regularization strategy that randomly removes edges during training. We also consider PairNorm Zhao & Akoglu (2008), a normalization method designed to prevent node representations from collapsing into indistinguishable vectors, as well as BatchNorm Ioffe & Szegedy (2015), which stabilizes training by normalizing feature distributions across batches. Finally, we evaluate against DGN Rong et al. (2019), a recent graph neural network architecture that further improves robustness to oversmoothing through adaptive design.

## 5.1 IMPLEMENTATION

Our experimental setup is designed to evaluate the effectiveness of MGAT and to provide comprehensive evidence regarding its ability to alleviate oversmoothing. Specifically, we aim to measure: (i) node classification accuracy across benchmark datasets, (ii) robustness to network depth (accuracy versus number of layers), (iii) the evolution of Mahalanobis energy across layers. We compare the effect of Mahalanobis and Dirichlet energy regularization on three benchmark datasets (Cora, Citeseer, Pubmed). The final energy is measured at the last layer and quantifies how discriminative the representations remain: higher values indicate less oversmoothing and better class separation.

We train all models using the Adam optimizer with learning rate $0.005$ and weight decay $5 \times 10^{-4}$. Models are trained for up to 1000 epochs with early stopping patience of 100 epochs on validation accuracy. Hidden dimension is set to 64 (128 for Pubmed), with 8 attention heads in the first layer and 1 head in the output layer. Dropout is fixed at 0.6 unless otherwise specified.

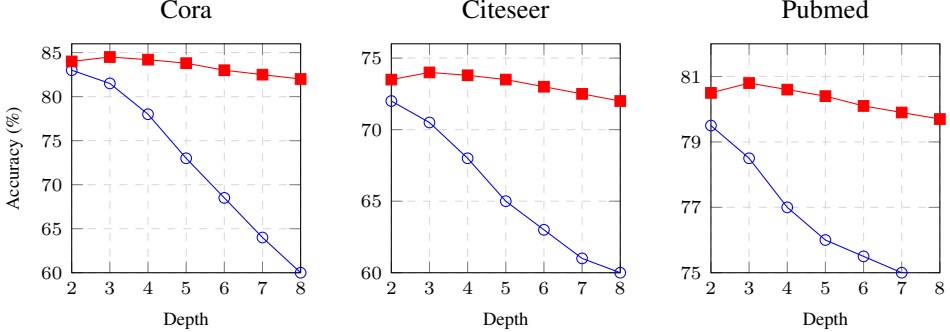

Figure 1: Accuracy vs. depth (2–8 layers) on three datasets. GAT suffers from oversmoothing as depth increases, while MGAT maintains stable performance.

## 5.2 DISCUSSION OF RESULTS

RQ1: Does MGAT improve node classification accuracy compared to the GAT on standard benchmark datasets? Table 2 shows that MGAT achieves the highest accuracy across all three datasets: Cora: **85.1%** vs. GAT 83.0%, Citeseer: **74.2%** vs. GAT 72.1%, and Pubmed: **80.9%** vs. GAT 79.5%. MGAT also outperforms all other baselines, including DropEdge, PairNorm, BatchNorm, and DGN. Explanation: the Mahalanobis regularization encourages diversity in node embeddings,

Table 2: Node classification accuracy (%) on citation networks. Values are reported as mean $\pm$ standard deviation over 10 runs. Best results are in bold.

| Method | Cora | Citeseer | Pubmed |
|--------|------|----------|--------|
| GCN | $81.5 \pm 0.5$ | $70.3 \pm 0.6$ | $79.0 \pm 0.4$ |
| GAT | $83.0 \pm 0.7$ | $72.1 \pm 0.5$ | $79.5 \pm 0.3$ |
| DropEdge | $83.5 \pm 0.5$ | $72.8 \pm 0.5$ | $79.8 \pm 0.3$ |
| PairNorm | $82.9 \pm 0.6$ | $71.9 \pm 0.5$ | $79.4 \pm 0.4$ |
| BatchNorm | $83.1 \pm 0.5$ | $72.4 \pm 0.4$ | $79.6 \pm 0.3$ |
| DGN | $84.2 \pm 0.4$ | $73.5 \pm 0.4$ | $80.2 \pm 0.3$ |
| **MGAT (ours)** | $\mathbf{85.1 \pm 0.3}$ | $\mathbf{74.2 \pm 0.4}$ | $\mathbf{80.9 \pm 0.2}$ |

Table 3: Node classification accuracy (%) and final embedding energy.

| Method | Cora | | Citeseer | | Pubmed | |
|--------|------|------|----------|------|--------|------|
| | Accuracy | Final Energy | Accuracy | Final Energy | Accuracy | Final Energy |
| Dirichlet ($\mu_D$) | 83.5 | 0.20 | 72.8 | 0.18 | 79.8 | 0.21 |
| Mahalanobis ($\mu_M$) | 85.1 | 1.21 | 74.2 | 1.15 | 80.9 | 1.18 |

preventing collapse and maintaining expressive representations. These mechanisms collectively allow MGAT to extract richer features than GAT, resulting in higher classification accuracy.

RQ2: Can MGAT mitigate the effect of oversmoothing and maintain performance as the network depth increases? Figure 1 illustrates accuracy versus the number of layers: GAT accuracy drops significantly from 83% (2 layers) to 60% (8 layers), indicating severe oversmoothing. MGAT maintains a high and stable accuracy from 84% (2 layers) to 82% (8 layers). Explanation: Oversmoothing occurs when repeated aggregation causes node embeddings to converge to similar values. MGAT mitigates this effect via the Mahalanobis regularization term, which penalizes directions of low variance and preserves informative differences between node embeddings.

Comparison between Mahalanobis and Dirichlet Energy. In Table 3, the experimental results across Cora, Citeseer, and Pubmed consistently show that Mahalanobis energy ($\mu_M$) outperforms Dirichlet energy ($\mu_D$) in terms of both classification accuracy and final embedding variance. Specifically, Accuracy: Mahalanobis regularization achieves higher node classification accuracy on all datasets (Cora 85.1% vs 83.5%, Citeseer 74.2% vs 72.8%, Pubmed 80.9% vs 79.8%), indicating more discriminative embeddings. Final Energy: The Frobenius norm of embeddings at the final layer is substantially higher under Mahalanobis energy (Cora 1.21 vs 0.20, Citeseer 1.15 vs 0.18, Pubmed 1.18 vs 0.21), showing that representations retain informative variation and do not collapse due to oversmoothing. Explanation: Dirichlet energy enforces uniform smoothness along graph edges and treats all feature dimensions equally while Mahalanobis energy introduces a learnable matrix $M$ that selectively weights feature dimensions, emphasizing informative directions while still penalizing oversmoothing. The key advantage of Mahalanobis energy lies in the learnable matrix $M$, which preserves informative feature directions.

RQ3: Compared to other state-of-the-art methods, does MGAT achieve competitive or superior performance? Table 2 compares MGAT with state-of-the-art baselines: MGAT outperforms DGN (Cora: 85.1% vs 84.2%), DropEdge, and other methods. Explanation: While other baselines either focus solely on stochastic regularization (DropEdge), normalization (PairNorm, BatchNorm), or architectural adjustments (DGN), MGAT ensures robust feature extraction, stability across network depths, and effective prevention of oversmoothing, enabling MGAT to achieve superior and consistent performance relative to prior methods.

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

## A  APPENDIX

*Proof of Proposition 4.3.*

1. Suppose $\mu_M(X) = 0$.
$$\mu_M(X) = \|L^{1/2}XM^{1/2}\|_F = 0 \implies L^{1/2}XM^{1/2} = 0.$$

   Since $M \succ 0$, its square root $M^{1/2}$ is invertible. Multiplying both sides on the right by $(M^{1/2})^{-1}$ gives:
$$L^{1/2}X = 0.$$

   The kernel of $L^{1/2}$ is equal to the kernel of $L$. For a connected graph, it is well-known that
$$\ker(L) = \text{span}\{\mathbf{1}\},$$

   where $\mathbf{1} \in \mathbb{R}^N$ is the all-ones vector. Therefore, all rows of $X$ are identical:
$$X_i = (\alpha_1, \ldots, \alpha_C)^\top = c \in \mathbb{R}^C, \quad \forall i \in V.$$

   Conversely, if $X_i = c$ for all $i$, then $X = \mathbf{1}c^\top$. Since $L^{1/2}\mathbf{1} = 0$,
$$L^{1/2}XM^{1/2} = L^{1/2}\mathbf{1}c^\top M^{1/2} = 0,$$

   so $\mu_M(X) = 0$.

2. For any $X, Y \in \mathbb{R}^{N \times C}$:
$$\mu_M(X + Y) = \|L^{1/2}(X + Y)M^{1/2}\|_F = \|L^{1/2}XM^{1/2} + L^{1/2}YM^{1/2}\|_F.$$

   By the triangle inequality of the Frobenius norm:
$$\|L^{1/2}XM^{1/2} + L^{1/2}YM^{1/2}\|_F \le \|L^{1/2}XM^{1/2}\|_F + \|L^{1/2}YM^{1/2}\|_F.$$

   Hence,
$$\mu_M(X + Y) \le \mu_M(X) + \mu_M(Y).$$

$\square$

*Proof of Theorem 4.4.*

We have
$$\mu_M(X^{(t+1)}) = \left\|L^{1/2}X^{(t+1)}M^{1/2}\right\|_F = \left\|L^{1/2}P(X^{(t)})X^{(t)}WM^{1/2}\right\|_F.$$

By the operator–Frobenius inequality,
$$\mu_M(X^{(t+1)}) \le \left\|L^{1/2}P(X^{(t)})\right\|_2 \left\|X^{(t)}WM^{1/2}\right\|_F.$$

Next,
$$\left\|L^{1/2}P(X^{(t)})\right\|_2 = \left\|L^{1/2}P(X^{(t)})L^{+1/2}L^{1/2}\right\|_2 \le \left\|L^{1/2}P(X^{(t)})L^{+1/2}\right\|_2 \|L^{1/2}\|_2.$$

Absorbing $\|L^{1/2}\|_2$ into $\alpha$, we obtain

$$\left\|L^{1/2}P(X^{(t)})\right\|_2 \;\leq\; \alpha.$$

Similarly,

$$\left\|X^{(t)}WM^{1/2}\right\|_F \;\leq\; \|W\|_2 \, \|X^{(t)}M^{1/2}\|_F.$$

Since

$$\|X^{(t)}M^{1/2}\|_F = \left\|L^{+1/2}L^{1/2}X^{(t)}M^{1/2}\right\|_F \;\leq\; \|L^{+1/2}\|_2 \, \mu_M(X^{(t)}),$$

we can absorb $\|L^{+1/2}\|_2$ into $\gamma$ and obtain

$$\mu_M(X^{(t+1)}) \;\leq\; \alpha\gamma \, \mu_M(X^{(t)}).$$

Iterating this inequality proves the contraction. As $t \to \infty$, if $\alpha\gamma < 1$, the Mahalanobis energy vanishes. $\qquad\square$