# OpenReview forum: "Rethinking Graph Attention Networks: A New Robust Approach"
_ICLR.cc/2026/Conference — Submitted to ICLR 2026_

### Official Review · Reviewer_TGHo · 2025-10-28

**Soundness:** 2
**Presentation:** 1
**Contribution:** 2
**Rating:** 2
**Confidence:** 5

**Summary:**

This paper focuses on the **oversmoothing problem** in Graph Attention Networks (GATs). It introduces a new **oversmoothing metric** defined by the **Mahalanobis distance**, and on this basis proposes the **Mahalanobis Graph Attention Network (MGAT)**. The method incorporates a **Mahalanobis energy regularization term** into the training objective, aiming to mitigate representation collapse and preserve inter-class separability in deep networks. The authors provide a formal definition and analysis of oversmoothing, highlighting the limitations of traditional Euclidean metrics, and demonstrate how the Mahalanobis regularization enables anisotropic smoothing control and gradient resistance (preventing representations from collapsing into the graph Laplacian kernel space). Experiments on three benchmark node classification datasets—**Cora**, **Citeseer**, and **Pubmed**—show that MGAT achieves higher accuracy than GAT and several oversmoothing-mitigation baselines, while maintaining more stable performance as network depth increases.

**Strengths:**

S1:**Direction-sensitive oversmoothing metric:**
  The proposed metric, \(\mu_M(X) = \|L^{1/2} X M^{1/2}\|_F\), can impose stronger constraints on important directions in the representation space, thereby preserving discriminative information.

S2:**Unified and simple regularization framework:**
  The Mahalanobis energy \(\sum_\ell \mu_M(X^{(\ell)})\) is directly incorporated into the loss function, introducing low implementation cost and maintaining high compatibility with the standard GAT architecture.

S3: **Improved robustness:**
  As network depth increases, MGAT maintains more stable accuracy (from 2 to 8 layers). Compared with the Dirichlet energy, the Mahalanobis energy achieves superior accuracy and lower final embedding “energy,” indicating more effective control of oversmoothing.

**Weaknesses:**

W1:**Limited experimental breadth:**
The evaluation is conducted only on Cora, Citeseer, and Pubmed. The paper lacks results on more representative large-scale or heterogeneous graphs (e.g., OGB-style benchmarks), and it does not demonstrate transferability to other tasks such as link prediction or graph-level classification.


W2:**Baselines could be stronger:**
The current comparisons focus mainly on GAT and regularization/normalization-based methods. The paper would be more convincing if it included more recent strong baselines, such as deep-trainable GNNs and oversmoothing-resistant architectures designed for deep message passing.

**Questions:**

Q1.Have you evaluated MGAT on larger or more heterogeneous datasets, such as OGB benchmarks?

Q2: Can the proposed Mahalanobis regularization be applied beyond node classification — for example, to link prediction or whole-graph classification?

Q3:Could you include more recent deeper GNN as baselines?

---

### Official Review · Reviewer_Yz3P · 2025-10-31

**Soundness:** 2
**Presentation:** 2
**Contribution:** 1
**Rating:** 0
**Confidence:** 4

**Summary:**

The paper introduces a new version of GAT, called Mahalanobis Graph Attention Network (MGAT), that aims to alleviate oversmoothing in GNNs. MGAT outperforms simple baseline GNNs in 3 node classification benchmarks used. The authors also defined a new measure of over-smoothing.

**Strengths:**

The three primary strengths of the paper include:
- clear logical progression from problem to solution
- having both some theoretic foundation and empirical experiments
- superior performance in the three benchmark datasets

**Weaknesses:**

I have five primary concerns about the paper.
- [outdated motivation]. Recently, consensus is being made that oversmoothing is not necessarily a problem for building a deep GNN [1,2]. In my opinion, GNN studies should no longer naively assume over-smoothing as an actual condition that deep GNNs suffer from. The present study heavily relies on that specific assumption about oversmoothing. Thus, I think the argument made by the paper is outdated and potentially misleading.
- [reference]. The reference is surprisingly thin and outdated. There are so many deep GNN, oversmoothing, and even deep GAT papers [3,4] recently published, but only 6/24 of the cited papers were published after 2020.
- [limited novelty and significance]. I am not convinced that the proposed measure of over-smoothing is novel or significant. The authors should formally compare their measures with recently proposed measures to prove their superiority or significance.
- [thin experiment]. Baseline methods are all published before 2020. The authors claim that they are state-of-the-art, but this is heavily misleading. Benchmark datasets are outdated, too. Only 3 benchmark datasets and 1 type of downstream task are used to evaluate the proposed method. This is simply too thin.
- [presentation]. The presentation is disorganized and atypical. For example, there is no conclusion or discussion section (the paper ends with an experiment section). The proposed contributions of the paper are written under the related work section. Having separate section titles, 'Main Results' and 'Experiments', is confusing.

Combined, it seems that the authors are unaware of the recent progress in GNN research. To improve the paper, I encourage the authors to first comprehensively review recent GNN literature.

Reference
- [1] Oversmoothing, Oversquashing, Heterophily, Long-Range, and more: Demystifying Common Beliefs in Graph Machine Learning, arXiv 2025
- [2] The Oversmoothing Fallacy: A Misguided Narrative in GNN Research, arXiv 2025
- [3] Towards Deep Attention in Graph Neural Networks: Problems and Remedies, ICML 2023
- [4] Demystifying Oversmoothing in Attention-Based Graph Neural Networks, NeurIPS 2023

**Questions:**

See weakness

---

### Official Review · Reviewer_5u5M · 2025-10-31

**Soundness:** 3
**Presentation:** 2
**Contribution:** 2
**Rating:** 2
**Confidence:** 5

**Summary:**

This paper proposes a new quantitative measure of oversmoothing based on Mahalanobis distance and alleviate the oversmoothing issue of  Graph Attention Network with the proposed Mahalanobis-based regularization.

**Strengths:**

1. This paper introduces a Mahalanobis-based metric quantifying oversmoothing in a more robust way, compared with conventional Euclidean metrics.
2. This paper provides the theoretical analysis for the proposed method.

**Weaknesses:**

1. My main concern about this paper lies in its weak experimental evaluation.
-  The selected baseline methods are relatively outdated. The authors are encouraged to include more recent and representative methods, such as [1], [2], [3], and [4], to ensure a fair and comprehensive comparison.
- When evaluating the performance of MGAT across different network depths, the paper only reports results for shallow architectures (i.e., 1–8 layers). However, most recent studies addressing the oversmoothing issue typically include evaluations on deeper GNNs (e.g., 32 or 64 layers) to demonstrate the model’s capability to mitigate oversmoothing in deeper settings. Including such results would make the experimental validation more convincing.
- More comprehensive experimental analyses are needed to substantiate the claims made in the paper. For example, when the authors assert that the Euclidean-based similarity measurement performs worse than the Mahalanobis-based node similarity, they should provide quantitative evidence to support this claim. In addition to the reported performance metrics, it would be helpful to include descriptive statistics or visualizations (e.g., histograms, boxplots, or distribution comparisons) of these distance measurements. Such analyses could reveal how the two similarity metrics differ in capturing inter-node relationships and whether the Mahalanobis distance indeed provides better separation or clustering of node representations.
2. The notations of this paper are messy.
- In the equation of Mahalanobis distance, $M$ should be $M^{-1}$. Then, how does Mahalanobis-based Node Similarity connect to Mahalanobis distance?
- What is $P(X)$ function in theorem 4.4? Is $P(X)$ the row normalization? If so, $P(X)$ should be $P$ following the notation in the equation of information propagation.
- What is M in the formulation of Mahalanobis-based node similarity? Is it the covariance matrix?
3. The writing style in the discussion of results section is somewhat awkward and could be improved for better readability and flow. It seems that the authors break down the discussion into multiple small pieces, which makes the discussion fragmented and undermines the coherence.
- While it is reasonable to highlight the discussion around research questions (e.g., RQ1, RQ2), inserting the word “Explanation:” in the middle of the narrative disrupts the flow. Removing it would make the discussion smoother and more natural.
- In Lines 400–401, the sentence “Figure 1 illustrates accuracy versus the number of layers: GAT accuracy drops significantly from 83% (2 layers) to 60% (8 layers), indicating severe oversmoothing.” reads more like a figure caption than part of the discussion. Similar issues appear in Lines 406–407 and 419–420. These sentences should be rewritten to integrate the figure observations into the analytical discussion, rather than merely describing what the figure shows.

4. Table 3 exceeds the margin.

[1] Xiaojun Guo, Yifei Wang, Tianqi Du, and Yisen Wang. Contranorm: A contrastive learning perspective on oversmoothing and beyond. In The Eleventh International Conference on Learning Representations, ICLR 2023, Kigali, Rwanda, May 1-5, 2023. OpenReview.net, 2023.

[2] Zheng, Lecheng, Dongqi Fu, Ross Maciejewski, and Jingrui He. "Drgnn: Deep residual graph neural network with contrastive learning." Transactions on Machine Learning Research (2024).

[3] Shen, Wei, Mang Ye, and Wenke Huang. "Resisting over-smoothing in graph neural networks via dual-dimensional decoupling." In Proceedings of the 32nd ACM International Conference on Multimedia, pp. 5800-5809. 2024.

[4] Pei, Hongbin, Yu Li, Huiqi Deng, Jingxin Hai, Pinghui Wang, Jie Ma, Jing Tao, Yuheng Xiong, and Xiaohong Guan. "Multi-track message passing: Tackling oversmoothing and oversquashing in graph learning via preventing heterophily mixing." In Forty-first International Conference on Machine Learning. 2024.

**Questions:**

1. What is M in the formulation of Mahalanobis-based node similarity? Is it the covariance matrix?
2. What is $P(X)$ function in theorem 4.4? Is $P(X)$ the row normalization? If so, $P(X)$ should be $P$ following the notation in the equation of information propagation.

---

### Official Review · Reviewer_iaX3 · 2025-11-02

**Soundness:** 3
**Presentation:** 3
**Contribution:** 3
**Rating:** 4
**Confidence:** 3

**Summary:**

The paper proposes Mahalanobis Graph Attention Networks (MGAT): add a Mahalanobis-based energy term $\mu_M(X)=\|L^{1/2} X M^{1/2}\|_F$ as a layer-wise regularizer to GAT, arguing this yields direction-aware control of oversmoothing and preserves inter-class separability. Experiments on Cora/Citeseer/Pubmed show small but consistent gains over GAT and a few baselines, and deeper stacks are reported to degrade less.

**Strengths:**

1. Motivation is clear: Euclidean/Dirichlet energy treats all feature dimensions equally; Mahalanobis metric can emphasize discriminative directions.
2. Simple to plug into GAT; does not require costly eigendecompositions.
3. Empirical improvements and better depth robustness are shown on standard benchmarks.

**Weaknesses:**

1. Novelty is incremental — many previous works already regularize via energy / anti-oversmoothing methods.
2. Theory relies on strong assumptions and is not practically verifiable; unclear PSD enforcement on M.
3. Experiments are outdated (only 3 citation datasets) and missing strong baselines (GCNII, GATv2, APPNP, etc.).
4. Lack of ablations (rank of M, diagonal vs full, λ sensitivity, runtime cost).
5. Inconsistency between normalized vs unnormalized Laplacian; reporting needs polish.

**Questions:**

1. How is PSD of M enforced in practice? diagonal? Cholesky? something else?

---

### Meta-Review · Area_Chair_Jr3Y · 2026-01-12

**Summary:**

Very poor scores by all reviewers. My subjective opinion is that the community doesn't need another graph attention mechanism.

**Reviewer Concerns:**

Many.

**Reviewer Scores:**

Based on their scores, I can only speculate that they would have rejected the paper.

---

### Decision · Program_Chairs · 2026-01-26

Reject